# Modeling Rare Human Disorders in Mice: The Finnish Disease Heritage

**DOI:** 10.3390/cells10113158

**Published:** 2021-11-13

**Authors:** Tomáš Zárybnický, Anne Heikkinen, Salla M. Kangas, Marika Karikoski, Guillermo Antonio Martínez-Nieto, Miia H. Salo, Johanna Uusimaa, Reetta Vuolteenaho, Reetta Hinttala, Petra Sipilä, Satu Kuure

**Affiliations:** 1Stem Cells and Metabolism Research Program, Faculty of Medicine, University of Helsinki, P.O. Box 63, 00014 Helsinki, Finland; tomas.zarybnicky@helsinki.fi; 2Biocenter Oulu, University of Oulu, P.O. Box 5000, 90014 Oulu, Finland; anne.heikkinen@oulu.fi (A.H.); salla.kangas@oulu.fi (S.M.K.); miia.h.salo@oulu.fi (M.H.S.); reetta.vuolteenaho@oulu.fi (R.V.); 3Oulu Center for Cell-Matrix Research, Faculty of Biochemistry and Molecular Medicine, University of Oulu, P.O. Box 8000, 90014 Oulu, Finland; 4PEDEGO Research Unit, University of Oulu, P.O. Box 8000, 90014 Oulu, Finland; johanna.uusimaa@oulu.fi; 5Medical Research Center, Oulu University Hospital, University of Oulu, P.O. Box 5000, 90014 Oulu, Finland; 6Research Centre for Integrative Physiology and Pharmacology, Institute of Biomedicine, University of Turku, 20520 Turku, Finland; Marika.Karikoski@utu.fi (M.K.); Guillermo.martineznieto@utu.fi (G.A.M.-N.); 7Turku Center for Disease Modelling (TCDM), Institute of Biomedicine, University of Turku, 20520 Turku, Finland; 8Clinic for Children and Adolescents, Division of Pediatric Neurology, Oulu University Hospital, P.O. Box 20, 90029 Oulu, Finland; 9GM-Unit, Laboratory Animal Center, Helsinki Institute of Life Science, University of Helsinki, 00790 Helsinki, Finland

**Keywords:** rare diseases, monogenic diseases, mouse models, CRISPR/Cas9, genome engineering, Finnish disease heritage

## Abstract

The modification of genes in animal models has evidently and comprehensively improved our knowledge on proteins and signaling pathways in human physiology and pathology. In this review, we discuss almost 40 monogenic rare diseases that are enriched in the Finnish population and defined as the Finnish disease heritage (FDH). We will highlight how gene-modified mouse models have greatly facilitated the understanding of the pathological manifestations of these diseases and how some of the diseases still lack proper models. We urge the establishment of subsequent international consortiums to cooperatively plan and carry out future human disease modeling strategies. Detailed information on disease mechanisms brings along broader understanding of the molecular pathways they act along both parallel and transverse to the proteins affected in rare diseases, therefore also aiding understanding of common disease pathologies.

## 1. Introduction

Understanding and treating human diseases requires thorough knowledge of disease-causing molecular and pathophysiological mechanisms. Despite the recent advances in induced pluripotent stem cell techniques, many of these aspects remain challenging to study, especially under physiological conditions in patient-derived material [1,2]. In order to be reliable and valuable, the disease model should recapitulate if not the entire human disease phenotype, then at least the key features of each specific disease under study [3]. For precision medicine, a good understanding of the genetic bases of variation in phenotypes and their interaction with the environment in health and disease are required [4,5]. Thus, animal models, and especially genetically modified (GM) mice, offer great potential to serve as precious preclinical models that facilitate basic understanding of disease pathomechanisms and provide clues for the development of treatment options and novel strategies to follow a treatment’s response.

### Mice as a Genetically Modified Model for Diseases

Mice have been used as an animal model in biomedical studies for decades and for many reasons, including their small size, efficient reproduction, relatively reasonable expenses and similarities in anatomy and physiology to humans [4]. The fact that mice are housed in controlled environment and that its genome was the first that was sequenced among the rodents makes it the quintessential and by far most extensively used animal employed in genetically modified models [6,7]. The possibility to derive and successfully culture mouse embryonic stem (ES) cells together with the importance of similarities between human and mouse genomes significantly contributed to the dominance of the mouse in genetic modeling [8,9].

Almost 15 years ago, an international knockout (KO) mouse consortium (IKMC) with the aim to inactivate all known mouse genes was established [10,11]. It has been extremely successful in generating and providing mouse gene inactivation models either in the form of targeted ES cells or mouse lines as, at least two thirds of the protein coding genes have been knocked out [12,13]. Together with the research community and systematic phenotyping conducted by the International Mouse Phenotyping Consortium (IMPC, https://www.mousephenotype.org, accessed on 15 September 2021), these models have produced a wealth of novel information on gene functions and facilitated the understanding of essential genetic requirements for life [12,14,15]. Detailed phenotyping of KO mice has also identified various full inactivation models that correlate or associate with Mendelian diseases [14,15]. However, straightforward full-gene inactivation appears to be less successful at modeling monogenic diseases than originally thought, and thus more precise models are urged.

One of the big challenges at hand is to model the approximately 6500 different human monogenic diseases in mice [16]. Although typically classified as rare diseases, many monogenic diseases are relatively common, either in general or in certain geographical regions and among certain ethnicities. In this review, we discuss a group of monogenic rare diseases enriched in the Finnish population and defined as the Finnish disease heritage (FDH) [17]. We will highlight how appropriate mouse models have greatly facilitated the understanding of disease manifestation in certain monogenic diseases and, on the other hand, emphasize the barriers in other diseases, as studies have been carried out mainly in cell culture models.

## 2. The Finnish Disease Heritage

The first scientific reports of FDH were published in early 1970s by Perheentupa et al. [18] and Norio et al. [19] who referred to a group of inherited clinical phenotypes that are, in proportion to population size, more common in Finland than elsewhere in the world. Today, 36 mostly autosomal recessive diseases (Table 1) are defined as FDH, varying in severity from embryonic lethal to adult-onset milder phenotypes [20]. The collection of monogenic FDH is constantly evolving (Table 1), as new diseases following the same patterns of enriched founder mutations are still being identified [21]. 

The enrichment of some monogenic diseases in Finland (especially in the north and east) was caused by geographical, linguistic and cultural isolation of this population in history [22]. The resultant characteristic features of population isolation (e.g., the founder effect, genetic drift and genetic isolation) have shaped the gene pool of Finns over the centuries, leading to the enrichment of certain disease-causing gene variants [23]. Although FDH diseases are rare, many of them share similarities with more common diseases, and this may sometimes delay a correct diagnosis. Thus, the importance of consultation of national expert centers that are linked to international networks is of ultimate importance for rare disease patients (https://ec.europa.eu/health/ern/, accessed on 1 September 2021). One example of such networks is European Reference Networks on rare diseases, which help professionals and expert centers to share knowledge on rare diseases requiring special care, serve as research and knowledge centers for treating patients from other EU countries, ensure the availability of treatment facilities and provide high-quality training for students and members of multidisciplinary teams.

### 2.1. State of the Art in Modeling the FDH in Mice

Thanks to the IKMC, many of the FDH genes have already been knocked out in mice (Table 1). However, only a few of the existing KO mouse lines recapitulate the full spectrum of FDH disease symptoms (Table 2). Moreover, as described in Table 2 for all FDH diseases, KO and rare transgenic (TG) and knock-in (KI) mouse models that have been generated for certain diseases selectively manifest only some of the disease features.

The selected FDH diseases and their animal model statuses are discussed in detail in the following section. These examples were chosen to highlight the great variation in disease manifestation and underline the importance of disease-specific knock-in mouse models for advancing knowledge on the molecular networks involved in normal development, physiology and disease pathogenesis. Specific attention was given to diseases with possible advances or suitability in preclinical treatments (AGU and HOGA), with severe developmental complications leading to devastating death of the fetus (HLS1), with severe infancy- or childhood-onset neurodegenerative storage disorders (Northern epilepsy and Salla disease) and with complex symptom spectrums (CHH and LCHAD). As exemplified below, strategies to generate and utilize in vivo models that faithfully genocopy their disease-specific variants and phenocopy major FDH disease symptoms are needed.

#### 2.1.1. Aspartylglucosaminuria (AGU)

AGU (OMIM 208400) represents one of the rare FDH diseases that has been subjected to some treatment strategies and preclinical development (see below). AGU is a recessive neurodegenerative disease which is characterized by progressive intellectual disability, skeletal and connective tissue abnormalities, behavioral changes (e.g., hyperactivity, tantrums and violence) and disruptive sleep patterns followed by premature death, usually before the age of 50 [95,96,97,98]. Developmental delay is the first typical sign of neurological defects, which become evident at 15–18 months of age. Children may manifest macrocephaly and early growth spurts in the infantile phase, but adult affected individuals end up having small brains and lower than average heights [99,100,101,102,103].

AGU is the most common autosomal recessive disease in Finland, and its prevalence is 1.7–5 per 100,000 live births [101,104]. AGU is caused by defects in the lysosomal aspartylglucosaminidase (AGA) enzyme. The AGU_Fin_ major variant, which consists of two nucleotide changes (c.482G > A and c.488G > C), covers 98% of cases in Finland, while the AGU_Fin_ minor variant (two base pairs deleted) is causative for 1.5% of the cases [105,106,107]. Although being enriched in Finland, AGU affects all ethnicities, and approximately 40 different AGA variants have been identified worldwide (A. Banning and R. Tikkanen, personal communication). Roughly half of the variants are missense mutations, while the rest represent many different aberrations without proper understanding of the genotype–phenotype causalities [98]. Notably, recent findings have demonstrated several patients with high residual AGA activity and a milder phenotype [108]. This is an important finding for development of treatment options, since it means that even less than 50% AGA activity (i.e., the carrier level) may result in a significant improvement of the phenotype.

The pathogenic AGA variants result in decreased AGA activity [109,110,111,112]. This causes a failure to break down the N-glycosidic bond of glycoproteins and leads to the progressive accumulation of AGA substrates, including glycoasparagine in lysosomes [113]. The KO mice of *Aga* recapitulated the biochemical defects of AGU, as glycoasparagine accumulation was evident in the studied tissues, but the model failed to manifest the early onset and progressive nature of specific disease symptoms [24,25].

Although there are no approved curative therapies currently available for AGU, some disease-modifying strategies have been and are currently being tested. Enzyme replacement therapy was shown to work in human cells and preclinical KO mouse models but has not been transferred to clinical trials, due to challenges in the required large-scale protein production and concerns for the blood–brain barrier transport [29,30,114,115]. Adenoviral gene therapy in *Aga* KO mice restored the enzyme activity [116,117]. The latest experiments with serotype-specific adeno-associated viral vectors safely targeting the central nervous system (AAV9) alleviated the neurological phenotype of *Aga* KO mice [26] and suggested clinical translatability due to good adherence with the dosage and blood–brain barrier transport requirements. Hematopoietic stem cell transplantation as a potential treatment of AGU showed promising results in the *Aga* KO mice but failed to improve the neuropsychologic and other clinical symptoms of AGU in clinical trials [114,115,118,119]. While promising, it is possible that previous enzyme replacement and gene and stem cell transplantation preclinical trials provoked unmerited hope due to the use of an inappropriate KO mouse model, which does not recapitulate the disease pathomechanisms caused by misfolded AGA protein.

Currently one AGU treatment option being tested is pharmacological chaperone therapy, which aims at facilitating proper folding of the mutant AGA protein. It can restore AGA activity in patient-derived fibroblasts and is currently in phase I or II clinical trials for efficacy and safety testing [108] (clinical trial). Of note, this clinical trial is conducted without preclinical animal testing because the *Aga* KO mouse is not a suitable model for this type of therapeutic strategy, and animal testing is not required for drug repurposing with approved agents such as anhydrous betaine (Cystadane), the chaperone used for AGU treatment, which is already approved for the treatment of homocystinuria. Important for future studies of targeted therapies in AGU is that a mouse model mimicking human disease-causing variant(s) is still lacking but would be highly desirable to facilitate drug development.

#### 2.1.2. Cartilage Hair Hypoplasia (CHH)

CHH (OMIM 250250), originally described by McKusick et al. in 1965, is metaphyseal chondrodysplasia characterized by a short stature, sparse hypoplastic hair and immune deficiency. Patients with CHH may also suffer from gastrointestinal dysfunctions, anemia and impaired spermatogenesis [120,121,122]. Especially due to malignancies and diseases of the respiratory system, patients with CHH have an increased risk of early mortality [123]. CHH is enriched among the Old Order Amish and Finnish populations, and the incidence in Finland is 1 in 23,000 births [124]. CHH is caused by defects in the long non-coding RNA gene known as *RMRP* [125]. The most common pathogenic variant is n.71A > G (previously known as n.70A > G) substitution, representing 92% of the disease-causing variants among Finnish CHH patients and also being the most frequent, if not the only one, in the Amish population [126,127].

*RMRP* encodes the RNA component of mitochondrial RNA processing endoribonuclease. It has well-characterized roles in many cancers, possibly due to its recently recognized function to inhibit p53 [128,129]. The *RMRP* KO mouse model indicates that it is essential for early embryonic development, as homozygous null mice die in utero [31]. Due to the missing mammalian CHH model, the pathophysiological mechanisms leading to the disease symptom spectrum remains poorly understood.

#### 2.1.3. Hydrolethalus Syndrome 1 (HLS1)

HLS1 (OMIM 236680) is a lethal disease with multiple developmental anomalies at the fetal stage, and it leads to stillbirth or death of the affected child soon after birth. The characteristic findings of HLS1 in the central nervous system are hydrocephalus, missing midline structures of the brain and a keyhole-shaped foramen magnum [130,131,132]. The incidence of this syndrome is 1 in 20,000 births in Finland [133,134,135]. Patients with similar or somewhat milder clinical features have also been reported outside Finland [133,136,137,138]. However, no causative variants have been confirmed in these cases.

An autosomal recessive variant of the *HYLS1* gene carrying the c.1416A > G transition has been identified as the disease-causing variant in HLS1. This A-to-G nucleotide transition in exon 6 leads to substitution of the conserved aspartate-211 to glycine (p.D211G) in the HYLS protein [139]. Hydrolethalus syndrome 2 (HLS2, OMIM 614120) has similar features to HLS1, but the causative variant resides in the *KIF7* gene [134]. Both HLS1 and HLS2 belong to primary ciliopathies, a diverse spectrum of neurodevelopmental disorders [135].

*HYLS1* encodes centriolar- and ciliogenesis-associated protein, which has been shown to play an important role in ciliogenesis in *Caenorhabditis elegans* and *Drosophila melanogaster* [140,141,142]. In vitro studies on human-derived cells and the examination of tissues from aborted HLS1 fetuses suggest that *HYLS1* is a transcriptional regulator essential for fetal development [143]. However, the exact cellular and molecular mechanisms behind severe malformations of the brain and other organs in HLS1 are yet unknown.

Currently, there are no publications on mouse models lacking *HYLS1* or replicating the disease-causing variant of HLS1. The severity of the syndrome and the utmost significance of HYLS1 protein to fetal development highlight the importance of further studies to advance our knowledge on molecular networks involved in embryo development both in health and disease.

#### 2.1.4. Hyperornithinemia with Gyrate Atrophy of the Choroid and Retina (HOGA)

HOGA (also known as GACR, OMIM 258870) is a rare autosomal recessive disorder characterized by progressive chorioretinal degeneration showing clinical symptoms during the first and second decades of life. It leads to visual impairment and blindness in adolescence or adulthood and early cataract formation, and it also often leads to neurological abnormalities and type II muscle fiber atrophy [144,145].

HOGA has an estimated global incidence of 1 in 1,500,000, being the highest in the Finnish population with an incidence of 1 in 50,000 [145,146]. This rare genetic disease is caused by defects in the ornithine-degrading mitochondrial enzyme, ornithine delta-aminotransferase (*OAT*, OMIM 613349), which leads to *hyper*ornithinemia in the plasma [144]. To date, more than 60 variants have been identified in the *OAT* gene that cause HOGA, 90% of which account for missense or frameshift changes [145]. From all the different variants, the C-terminal domain leucine-402 to the proline variant (L402P) and the catalytic site variant arginine-180 to the threonine (R180T) are the most frequent ones among the Finnish population [145].

The OAT enzyme function is bidirectional; in the early neonatal period it is more active in ornithine production, but later, this is reversed. Therefore, the adolescence-onset clinical HOGA manifestation can be delayed by a rigorous arginine-restricted diet, since this amino acid is the main source of ornithine. Several of the human variants have only been studied in patient-derived cells, human cell lines or in yeast [147,148,149]. In addition, there are two mouse models available for studying the function of the *Oat* and HOGA disease: a full KO line (*Oat*^tm1Dva^) and a mouse line with a spontaneous recessive mutation, called retarded hair growth (*Oat*^rhg^), that harbors glycine-353 to alanine (G353A) substitution in OAT protein [150]. Studies with KO mice showed that the loss of *Oat* leads to neonatal lethality due to *hypo*ornithinemia and subsequent arginine synthesis failure in the small intestine [66]. Notably, neonatal lethality has not been observed in HOGA patients, although one asymptomatic individual was reported to have transient neonatal hypoornithinemia. This suggests that in human infants, the biochemical phenotype might be similar, albeit milder than in KO mice. The neonatal lethality observed in KO mice can be rescued by arginine supplementation, and adult mice develop *hyper*ornithinemia and retinal degeneration comparable to humans. The *Oat*^rhg^ mice model the classical gyrate atrophy disease with chorioretinal deterioration and *hyper*ornithinemia [151].

Despite the existence of two mouse models, the HOGA pathogenesis and the exact molecular mechanism of the different OAT variants remain elusive, which reflects the current lack of treatments. This highlights the necessity for generating animal models that recapitulate the human variants to facilitate understanding of the HOGA disease and the development of efficient therapies.

#### 2.1.5. Northern Epilepsy (EPMR)

Northern epilepsy (progressive epilepsy with mental retardation (EPMR), OMIM 610003) is a neurodegenerative storage disease and a form of neuronal ceroid lipofuscinosis (NCL) [152]. The disease onset is at 5–10 years of age and marked by the appearance of generalized tonic-clonic seizures. The frequency of epileptic seizures increases toward puberty, after which the epileptic activity decreases. Mental deterioration is typically observed 2–5 years after the onset of epilepsy, and it is progressive despite the decline in the frequency of epileptic seizures toward adulthood. Other features associated with Northern epilepsy are pubertal behavioral difficulties as well as problems in fine motor tasks and equilibrium [153]. Progressive brain atrophy [147] and the accumulation of lipopigment in the cytoplasm of neurons and other cell types are observed at the tissue level [152].

Northern epilepsy is caused by a missense variant (c.70C > G) in the *CLN8* gene encoding the CLN8 transmembrane ER and ERGIC protein [154], which is involved in the transport of newly synthesized lysosomal enzymes from the endoplasmic reticulum (ER) to the Golgi apparatus [148]. This pathogenic variant results in arginine to glycine substitution at amino acid 24 (R24G) of CLN8 [154]. While the variant causing Northern epilepsy has only been found in the Finnish population, other pathogenic variants in CLN8 have been found elsewhere with different NCL disease phenotypes (OMIM 600143). In addition to its role in the ER for Golgi trafficking of lysosomal enzymes, CLN8 has recently been suggested to function in the regulation of endo-lysosomal dynamics and dendritic morphology [149].

Motor neuron degeneration (*mnd*) in mice, caused by spontaneous mutation in *Cln8*, is the most characterized murine model for CLN8 deficiency. This naturally occurring variant (c.267–268insC at codon 90 of *Cln8*) results in frameshift mutation and a premature stop codon [154]. While the Cln8*^mnd^*^/*mnd*^ mouse recapitulates many NCL features [57,155], the genotype, intracellular targeting and some of the phenotypic features do not correspond to those observed specifically in Northern epilepsy [156,157,158]. Recently, AAV9 gene therapy was tested in *Cln8^mnd^* mice, with promising NCL-alleviating results [58]. To date, no specific KI models of Northern epilepsy have been published.

#### 2.1.6. Salla Disease (SD)

Salla disease (OMIM 604369) is a slowly progressing neurodegenerative lysosomal sialic acid storage disorder. Its first clinical signs, such as muscular hypotonia, ataxia, transient nystagmus and retarded motor development, are usually observed during the first year of life. The life expectancy of the patients is considered to have slightly decreased. The severity of the clinical manifestations may vary, but all patients are intellectually disabled [159], and characteristic MRI findings include delayed myelination in the brain [160]. Salla disease manifests with the accumulation of sialic acid in lysosomes due to deficient sialic acid transportation out of the lysosomal membrane, and one of the key clinical characteristics of the patients includes excessive amounts of secreted sialic acid in their urine [159].

Salla disease and other related and often more severe sialic acid storage diseases (SASD) are caused by pathogenic variants in the SLC17A5 protein, which functions as a sialic acid transporter in lysosomes [161]. Most of the Finnish Salla disease patients are homozygous for missense variant R39C [162]. SLC17A5 also has nitrate transporter activity [156], and the extralysosomal localization in the CNS indicates that it may have other functional roles in addition to lysosomal sialic acid transport [163].

The *SLC17A5* gene is highly conserved across species. The mouse *Slc17A5* sequence is an 86.26% match with its human ortholog (Ensembl release 104, May 2021 [164]). The *Slc17A5* KO mouse model has a severe hypomyelination phenotype leading to death after 3 weeks of life [89,90]. Even though the KO phenotype in mice resembles a human disease in the terms of hypomyelination and lysosomal accumulation, the short life span and total loss of the protein do not facilitate studies of the disease mechanisms that could be causative for human patients. Different SLC17A5 variants cause different disease phenotypes, varying from mild to severe due to the amount of residual functional activity [162]. Interestingly, SLC17A5 variants have also been recognized as candidates for Parkinson’s disease susceptibility genes [165]. Therefore, specific KI mouse models that recapitulate the phenotype of Salla disease are needed to better understand the versatile functions of the SLC17A5 protein and pathogenetic processes behind the phenotypic spectrum.

#### 2.1.7. Evolving FDH: An Example of Long-Chain 3-hydroxyacyl-CoA Dehydrogenase Deficiency (LCHAD)

Although the definition for FDH has remained the same for decades, new diseases have been identified and are constantly included in this definition. One of the diseases currently under consideration for FDH is long-chain 3-hydroxyacyl-CoA dehydrogenase deficiency (LCHAD, OMIM 609016), which is a mitochondrial disorder of long-chain fatty acid oxidation characterized by infancy- or early childhood-onset metabolic acidosis, hypoketotic hypoglycemia, hypotonia, liver dysfunction, cardiomyopathy and arrhythmias [157,158]. Other symptoms of LCHAD deficiency include later-onset chronic peripheral neuropathy and pigmentary retinopathy. Furthermore, LCHAD deficiency carriers have an increased risk of pregnancy complications, including acute fatty liver; hemolysis, elevated liver enzymes and low platelets (HELLP) syndrome; and pre-eclampsia [166,167,168]. Treatment of LCHAD deficiency comprises a low-fat and high-carbohydrate diet, as well as avoidance of fasting. To obviate the fasting period during the nighttime in infants and small children, a nasogastric tube or gastrostomy is beneficial. During metabolic stress, like infections such as gastroenteritis, an infusion of intravenous glucose is used [169].

LCHAD deficiency in Finland is typically caused by the c.1528G > C founder mutation in the hydroxyacyl-coenzyme A dehydrogenase trifunctional multienzyme complex subunit alpha (*HADHA*) gene encoding the α-subunit of the mitochondrial trifunctional protein (MPT) complex. The missense variant leads to the substitution of glutamate-510 to glutamine (E510Q) [170]. This results in the accumulation of long-chain 3-hydroxy fatty acids and long-chain 3-hydroxyacylcarnitines in the patients’ tissues, but the damaging mechanisms causing the spectrum of symptoms manifesting at very different stages of life are not fully understood [171,172]. Early studies suggested that defects in mitochondrial energy metabolism might underlay the skeletal muscle defects, and this has been supported by in vitro studies with mitochondria isolated from rat skeletal muscle [173,174,175]. Homozygote *Hadha* KO mice with exon 15 deletion are reported to result in either early postnatal or embryonic lethality, while heterozygosity leads to hepatic steatosis at a young age (3 months) and hepatocellular carcinoma without cirrhosis at an older age (>13 months) [70,176].

Although dietary restrictions such as prevention of fasting and supplementation of carbohydrates and medium-chain triacylglycerols, together with acute infection avoidance, form the basis of LCHAD deficiency management, they insufficiently protect patients from long-term adverse effects. Better in vivo understanding of the consequences of metabolite accumulation in different tissues is needed to improve the life quality and expectancy of LCHAD deficiency patients.

## 3. Conclusions

For practical and ethical reasons, model organisms have been used as simplified models of humans to study the genetic, molecular and physiological basis of complex traits and to find therapeutic targets for human diseases [4]. At the same time, the use of animal experiments is ethically controversial and needs to be thoroughly justified. Non-animal approaches, based mainly on cell or tissue culture and in silico computational methods, may help to reduce the number of animals used for experimentation and predict clinical outcomes in a limited manner. Many rare hereditary diseases have a multi-organic clinical presentation. Cell and organ culture or computational models are incapable of modeling such biological complexity. Thus, deciphering the physiological functions, pathological processes and interactions between tissues necessitates the use of the whole organism. Consequently, mouse models continue to have a crucial role in biomedical research as well as drug discovery and development [177].

Although traditional KO mice have been valuable tools for studying gene functions, in many cases, the use of full inactivation of gene function has not successfully modeled the human monogenic diseases caused by point mutations in the corresponding gene (Table 2). The accurate recapitulation of disease-causing variants in mice could be the key to providing valuable tools for studying rare diseases. In recent years, the clustered and regularly interspaced palindromic sequences and CAS9 endonuclease (CRISPR/Cas9) genome editing technique have significantly improved the efficiency of generating animal models and been proven especially useful for the generation of point mutation models. In addition, CRISPR/Cas9 technology may alleviate some of the ethical concerns of animal use, especially during model creation. The three Rs principle—replacement, reduction and refinement [178]—could be tackled in several ways when using the CRISPR/Cas9 method, which increases the precision in genome editing and may help to refine and reduce mice, especially when generating complex models or engineering a mutation to the existing mutant background [179]. 

Importantly, in addition to providing valuable tools for diagnostic, prognostic and therapeutic strategies for rare diseases, precision mouse models facilitate our understanding of the pathomechanisms in the more common diseases affecting the same signaling pathways or cell and tissue types. A good example of this is a rare inherited Tangier disease, named after the location in which it was initially discovered and characterized by significantly reduced levels of high-density lipoproteins (HDLs) in the blood. Tangier disease is caused by a mutation in the *ABCA1* gene encoding the ATP-binding cassette transporter A1 (ABCA1), leading to impaired cholesterol efflux capacity. Research on Tangier disease and ABCA1 has had a tremendous impact on the understanding of HDL cholesterol metabolism and atherosclerosis [180,181]. Furthermore, the involvement of ABCA1 has been demonstrated in the pathophysiology of Alzheimer’s disease, for which it is now studied as a novel therapeutic target [182].

Owing to pioneer work by clinicians and geneticists working with FDH patients, the clinical implications and the genetic etiologies of the FDH are well known. However, the consequences of pathogenic variants and their contribution to disease progression at the molecular, cellular and tissue levels remain to be resolved for many diseases. CRISPR/Cas9 technology now allows the generation of disease models for the FDH with exact correspondence of disease-causing human variants in the mouse genome. From a total of 36 FDH diseases, 8 completely lack a mouse model, and in 14 diseases with the existing mouse model(s), only part of the disease symptoms is recapitulated (Table 1 and Table 2). In 2020, the corresponding authors of this review established a national FinnDisMice research consortium with the aim of generating mouse models that recapitulate the human disease-causing variants of the equivalent disease. The overall goal of this effort is to facilitate understanding of the disease pathomechanisms, provide preclinical tools for the development of novel treatment strategies and increase understanding of the molecular mechanisms behind similar common diseases, such as amyotrophic lateral sclerosis (ALS), Parkinson’s disease and other degenerative disorders.

Future technical developments in genome engineering will likely simplify the generation of humanized mouse models, with whole mouse genes substituted with human orthologs with or without disease-causing mutations [183]. Humanized models would be especially useful in the cases where there is low homology between the mouse and human orthologs, as well as in therapeutic development. Humanizing whole genes can be performed either by replacing the mouse gene with the human protein coding region and intervening introns or by excluding some or all the introns to reduce the size of the genomic fragment to be inserted. However, the latter option can result in unexpected surprises, such as in splicing [183]. Thus, when humanizing entire genes, the size of the genomic fragment inserted tends to be dozens or even hundreds of kilobases. The traditional targeting via homologous recombination in mouse ES cells is very inefficient, especially for inserting such long sequences. CRISPR/Cas9 in combination with the vectors accepting large genomic inserts (bacterial artificial chromosomes (BACs)) or single-stranded oligodeoxynucleotides has been proven to improve the targeted insertion of human *TERT* and *SIRPA* genes [184,185]. With the use of CRISPR/Cas9, the generation of humanized mice has become easier; however, it is still far from routine work. The current and future developments in the technique are likely to provide easier and more efficient ways of humanizing mice. While waiting for those technical advances, point mutation models of monogenic diseases provide a great preclinical tool for a wide research community.

Interestingly, a recent work with so-called “wildling” mice revealed that C57Bl/6 laboratory mice, which represent a much-used wild type strain where genetic engineering is carried out, with natural microbiota and pathogens phenocopy human immune responses better than the normally used pathogen-free C57Bl/6 mice [186]. Wildling mice were generated by transferring C57Bl/6NTac embryos into the oviducts of wild female mice. The resulting wildling mice had bacterial microbiomes in the gut, skin and vagina similar to those of wild mice, and the microbiota remained stable during several further generations (tested until F5 generation). Using a similar strategy in genetic manipulations could provide better translational value for precision medicine mouse models, especially in the cases where disease pathomechanisms involve the immune system (CHH) or neurodegeneration (Northern epilepsy and Salla disease).

Animal studies remain indispensable for understanding the complex biology and physiology of living organisms, and many regulatory authorities require them as a safety checkpoint in testing new treatments, whether based on drugs, genetic solutions or regenerative processes. Additionally, animal-based research often plays a crucial role in providing new insight into the understanding of diseases, associated pathologies and the identification of targets to which the treatment is directed [6]. Thus, animal experiments remain the best—but yet imperfect—model to predict and characterize human diseases [187].

## Figures and Tables

**Table 1 cells-10-03158-t001:** Finnish disease heritage diseases and the affected genes with the major Finnish mutation.

Disease	Gene	Major Mutation	Mouse Model Recapitulating Disease
Aspartylglucosaminuria (**AGU**)	Aspartylglucosaminidase (***AGA***)	c.488G > C p.C163S	*Aga* KO **+**
Autoimmune polyendocrinopathy -candidiasis-ectodermal dystrophy (**APECED**)	Autoimmune regulator (**AIRE**)	c.769C > T p.R257X	*Aire* KO **+/−**
Cartilage hair hypoplasia (**CHH**)	RNA component of mitochondrial RNA processing endoribonuclease (***RMRP***)	n.71A > G	*Rmrp* KO **−**
Choroideremia (**CHM**)	Rab escort protein 1 (***REP1***)	c.1639 + 2insT	*Rep1 c*KO **+**
Lactase deficiency, congenital (**CLD**)	Lactase (***LCT***)	c.4170T > A p.Y1390X	**N/A**
Ceroid lipofuscinosis, neuronal, 1 (**CLN1**)	Palmitoyl-protein thioesterase 1 (***PPT1***)	c.364A > T p.R122W	*Ppt1* KO **+**
Ceroid lipofuscinosis, neuronal, 3 (**CLN3**)	CLN3, battenin (***CLN3***)	g.462-677del p.G154Afs*29	*Cln3* KO **+/−** *Cln3*(Δex7/8) KI **+/****−**
Ceroid lipofuscinosis, neuronal, 5 (**CLN5**)	CLN5-intracellular trafficking protein (***CLN5***)	c.1175_1176delAT p.Tyr392*	*Cln5 KO* **+/−**
Cornea plana 2 (Cornea plana congenital, **CNA2**)	Keratocan (***KERA***)	c.740A > G p.N247S	*Kera KO* **+/−**
Finnish congenital nephrosis (**CNF**)	Nephrin (***NPHS1***)	c.121_122delCT p.R1109X	*Nphs1* KO **+**
Cohen syndrome (**COH1**)	Vacuolar protein sorting 13 homolog B (***VPS13B***)	c.3348_3349delCT p.C1117fs	**N/A**
Diarrhea, secretory chloride, congenital (**DIAR1**)	Solute carrier family 26, member 3 (***SLC26A3***)	c.-26 + 2T > C p.V317del	*Slc26a3* KO **+/−**
Diastrophic dysplasia (**DTD**)	Solute carrier family 26 member 2 (***SLC26A2***)	c.-26 + 2T > C	*Slc26a2* KI **+**
Epilepsy, progressive myoclonic, 1 (**EPM1**)	Cystatin B (***CSTB***)	12 nucleotide expansion in promoter	*Cstb* KO **+**
Epilepsy, progressive, with mental retardation (**EPMR**)	CLN8 transmembrane ER and ERGIC protein (***CLN8***)	c.70C > G p.R24G	*Cln8* ^mnd^ **+/−**
Amyloidosis, Finnish type (**FAF**)	Gelsolin (***GSN***)	c.654G > A p.D187N	*hGSN* Tg **+/−**
Glycine encephalopathy (**GCE**)	Glycine decarboxylase (***GLDC***)	c.1691G > T p.S564I	*Gldc* KO **+**
Gracile syndrome (**GRACILE**)	BCS1 homolog, ubiquinol-cytochrome c reductase complex chaperone (***BCS1L***)	c.232A > G p.S78G	*Bcs1l KI* **+**
Hydrolethalus syndrome 1 (**HLS1**)	HYLS1 centriolar and ciliogenesis associated (***HYLS1***)	c.1416A > G p.D211G	**N/A**
Hyperornithinemia with gyrate atrophy of choroid and retina (**HOGA**)	Ornithine aminotransferase (***OAT***)	c.1205T > C p.L402P	*Oat* KO **+/−**
Imerslund-Grasbeck syndrome 1 (**IGS1**)	Cubilin (***CUBN***)	c.3891G > A p.P1297L	*Cubn* KO **−**
Infantile onset spinocerebellar ataxia (**IOSCA**)	C10ORF2-chromosome 10 open reading frame 2 (***C10ORF2***)	c.1708A > G p.Y508C	*C10orf2* KI **+**
Lethal arthrogryposis with anterior horn cell disease (**LAAHD**)	RNA transport mediator (***GLE1***)	c.432-10A > G p.T144_E145insPFQ	**N/A**
		Compound heterozygote	
Lethal congenital contracture syndrome 1 (**LCCS1**)	RNA transport mediator (***GLE1***)	c.432-10A > G p.T144_E145insPFQ	**N/A**
Long-chain 3-hydroxyacyl-CoA dehydrogenase (**LCHAD**) deficiency *****	Hydroxyacyl-coenzyme A dehydrogenase trifunctional multienzyme complex subunit alpha (***HADHA***)	c.1528G > C p.E510Q	*Hadha* KO **+/−**
Lysinuric protein intolerance (**LPI**)	Solute carrier family 7 member 7 (***SLC7A7***)	c.895-2A > T p.T299IfsX128	*Slc7a7* KO1 **+** *Slc7a7* KO2 **−**
Muscular dystrophy-dystroglycanopathy (congenital with brain and eye anomalies), type a, 3 (**MDDGA3**)	Protein O-linked mannose N-acetylglucosaminyltransferase 1 (beta 1,2-) (***POMGNT1***)	c.1539+1G > A p.L472_H513del	*Pomgnt1* KO1 **+** *Pomgnt1* KO2 **+/****−**
Meckel syndrome type 1 (**MKS1**)	MKS transition zone complex subunit 1 (***MKS1***)	c.1408-7_35del p.G470fs	*Mks1* KO **+**
Mulibrey nanism (**MUL**)	Tripartite motif containing 37 (***TRIM37***)	c.493-2A > G p.R166fs	*Trim37* KO **+**
Ovarian dysgenesis 1 (**ODG1**)	Follicle stimulating hormone receptor (***FSHR***)	c.566C > T p.A189V	*Fshr* KO **+**
Progressive encephalopathy with edema, hypsarrhythmia and optic atrophy (**PEHO**)	Zinc finger HIT-type containing 3 (***ZNHIT3***)	c.92C > T p.S31L	**N/A**
Polycystic lipomembranous osteodysplasia with sclerosing leukoencephalopathy 1 (**PLOSL1**)	Transmembrane immune signaling adaptor TYROBP (***TYROBP***)	Ex1-4del: 5,3 kb deletion	*Tyrobp* KO **+**
**RAPADILINO** syndrome	RecQ like helicase 4 (***RECQL4***)	c.1390+2delT	*Recql4 KO1-3* **+/−**
Retinoschisis (**RS1**)	Retinoschisin 1 (***RS1***)	c.214G > A p.E72K	*Rs1* KO **+/−**
Salla disease (**SD**)	Solute carrier family 17 member 5 (***SLC17A5***)	c.115C > T p.R39C	*Slc17a5* KO **+/−**
Tibial muscular dystrophy (**TMD**)	Titin (***TTN***)	11-bp change in the last exon	*Ttn* cKO **+/−**
		Glu → Val	
		Val → Lys	
		Thr → Glu	
		Trp → Lys	
Usher syndrome, type III (**USH3**)	Clarin (***CLRN1***)	c.528T > G p.Tyr176Ter	*Clrn1* KO **+/−**

* Currently under consideration for FDH. c. = coding DNA variant; cKO = conditional knockout; KI = knock-in; KO = knockout; n. = non-coding DNA variant; p. = protein variant; + = mouse model(s) exists and recapitulates the majority of the human disease symptoms; +/− = mouse model(s) exists and recapitulates some of the disease symptoms; N/A = mouse model does not exist or recapitulate disease symptoms.

**Table 2 cells-10-03158-t002:** Description of characteristic clinical features of FDH diseases and respective studies in possible knockout (KO), transgenic (TG) or knock-in (KI) mouse model(s).

Disease	OMIM	Disease Manifestation	Mouse Model	Model Utility
Aspartyl- glucosaminuria (**AGU**)	208,400	Lysosomal storage disease with infantile growth spurt, progressive mental retardation in children, abnormalities in the central nervous system and skeleton and connective tissue lesions	***Aga*****KO:** recapitulates well lysosomal storage disease, but manifests symptoms only in aged animals [24,25]	Enzyme replacement and gene therapies successfully used in KO model [26,27,28]
Autoimmune polyendocrinopathy, candidiasis and ectodermal dystrophy (**APECED**)	240,300	Multi-symptomatic endocrinopathy with fungal infections and ectodermal changes	***Aire* KO**: recapitulates autoimmune symptoms of APECED [29,30]	
Cartilage hair hypoplasia (**CHH**)	250,250	Metaphyseal chondrodysplasia, short stature, sparse hair, immune deficiency, gastrointestinal dysfunctions, anemia, increased risk for lymphoma and impaired spermatogenesis	***Rmrp*****KO**: embryonic lethal [31]	
Choroideremia (**CHM**)	303,100	X-linked progressive degeneration of the retinal pigment epithelium, photoreceptors and choroid leading to vision loss of affected males	***Rep1 c*****KO:** conditional knockouts showed the early onset and progressive retinal degeneration, patchy depigmentation of the retinal pigment epithelium and Rab prenylation defects, leading to premature accumulation of deposits in retinal pigment epithelium [32,33]	
Lactase deficiency, congenital (**CLD**)	223,000	Infantile-onset severe diarrhea and failure to thrive	N/A	
Ceroid lipofuscinosis, neuronal, 1 (**CLN1**)	256,730	Infantile-onset, lethal neurodegenerative disease leading to psychomotor deterioration, muscular hypotonia, ataxia, myoclonia, microcephaly, progressive epilepsy and visual impairment causing blindness	***Ppt1*****KO:** (exon 9 or exon 4): similar CLN1-like phenotypes with blindness, seizures and myoclonic jerks; progressive motor difficulties leading to hind limb paralysis and death [34,35,36,37,38,39]	Gene therapy testing conducted using the *Ppt1* KO mice [24,38,40]
Ceroid lipofuscinosis, neuronal, 3 (**CLN3**)	204,200	Fatal neurodegenerative disorder with childhood-onset vision impairment, intellectual disability, movement problems, speech difficulties and seizures, which worsen over time	***Cln3*****KO:** neuronal storage disorder and other neuropathologies [41] ***Cln3*(Δex7/8) KI:** degenerative changes in retina, cerebral cortex and cerebellum; neurological deficits and premature death [40,42]	Although both mouse models recapitulate the aspects of CLN3, they either also show non-neuronal or genetic background-dependent phenotypes, thus not being good models for interventional studies [40,43]
Ceroid lipofuscinosis, neuronal, 5 (**CLN5**)	256,731	Childhood-onset developmental regression, myoclonic epilepsy, ataxia, vision loss, speech problems and a decline in intellectual function with varied life expectancy	***Cln5 KO:*** progressive pathology of the brain mimics the CLN5 symptoms, and *Cln5* deficiency leads to microglial activation, defective myelination and changes in lipid metabolism [44,45]	
Cornea plana 2 (cornea plana congenital) (**CNA2**)	217,300	Congenital visual impairment, reduced curvature and hazy limbus of the cornea, opacities in the corneal stroma and marked corneal arcus at early age	***Kera KO:*** structural alterations recapitulate disease phenotype, but corneal transparency is normal [46]	
Finnish congenital nephrosis (**CNF**)	256,300	Prenatal onset of massive proteinuria, severe steroid- resistant nephrotic syndrome at birth and rapid progression to end-stage renal failure	***Nphs1*****KO:** severe proteinuria associated with kidney defects and leading to postnatal lethality [47,48]	
Cohen syndrome (**COH1**)	216,550	Non-progressive psychomotor retardation and microcephaly, characteristic facial features, retinal dystrophy, cardiac dysfunction, hyperlaxity of joints and intermittent neutropenia	***Vsp13b* KO (IMPC)**	
Diarrhea, secretory chloride, congenital (**DIAR1**)	214,700	Fetal-onset watery diarrhea, polyhydramnion and chronic diarrhea due to chloride absorption defect	***Slc26a3*****KO:** inpenetrant postnatal lethality, and survivors suffer from growth retardation and acidic chloridorrhea [49]	
Diastrophic dysplasia (**DTD**)	222,600	Chondrodysplasia causing severe growth retardation and structural and functional abnormalities of joints	***Slc26a2*****KI** (hypomorph): recapitulates essential aspects of DTD such as growth retardation, skeletal dysplasia and joint contractures [50]	Therapeutic approaches to improve skeletal deformity and short stature in DTD successfully tested using *Slc26a2* KI mice [51]
Epilepsy, progressive myoclonic, 1 (**EPM1**)	254,800	Childhood- or juvenile-onset progressive myoclonic epilepsy with variable severity	***Cstb*****KO:** phenocopies progressive ataxia and myoclonic seizures [52,53]	
Epilepsy, progressive, with mental retardation (**EPMR**)	610,003	A neurodegenerative, lysosomal storage disease characterized by childhood-onset epilepsy and progressive mental retardation	***Cln8*****^mnd^** (267–268insC; frameshift, predicted truncated protein): early onset retinal degeneration and adult-onset hindlimb weakness and ataxia, progressing to spastic paralysis of all limbs and death by 9–14 months; accumulation of intracytoplasmic and lipopigment immunoreactive to ATP synthase subunit c [54,55,56,57] ***Cln8*** **KO (IMPC)**	Gene therapy testing conducted using *Cln8^mnd^* mice [58]
Amyloidosis, Finnish type (**FAF**)	105,120	Amyloidogenic disease characterized by lattice corneal dystrophy, cranial neuropathy, bulbar signs, and dermatologic changes. Peripheral neuropathy and renal failure are less common symptoms	***hGSN*****Tg:** transgenic line expressing human D187N gelsolin modeling the pathogenic endoproteolytic cascade that leads to gelsolin amyloidogenic peptides and accumulation with amyloidogenesis is restricted to muscle tissue [59]	Mouse model was used to test D187N gelsolin-targeting nanobodies with positive results [60]
Glycine encephalopathy (**GCE**)	605,899	Accumulation of glycine in neonates. Disease varies from attenuated to fatal form and presents with lethargy, hypotonia, myoclonic jerks and apneas	***Gldc*****KO:** neonatal disease features with increased glycine levels, premature lethality and hydrocephalus, in addition to neural tube defects [61]	Abnormalities of folate metabolism and hydrocephalus were prevented by maternal supplementation of carbon donor to normalize folate cycle [62,63]
Gracile syndrome (**GRACILE**)	603,358	A mitochondrial disease characterized by severe growth retardation, lactic acidosis, nonspecific amino aciduria, cholestasis and abnormalities in iron metabolism, resulting neonatal or early infancy lethality	***Bcs1l KI***: similar phenotype to human diseases such as growth restriction (>4 wk), progressive liver disease, renal tubulopathy and premature death (<6 wk) [64,65]	
Hydrolethalus syndrome 1 (**HLS1**)	236,680	A lethal condition of fetus with hydramnion and multiple developmental anomalies, including central nervous system malformation, micrognathia, polydactyly, congenital heart defects and abnormal lung lobuli	***Hyls1* KO (IMPC)**	
Hyperornithinemia with gyrate atrophy of the choroid and retina (**HOGA**)	258,870	Hyperornithinemia presumably due to OAT deficiency; triad of progressive chorioretinal degeneration, early cataract formation and type II muscle fiber atrophy; progressive vision loss	***Oat*****KO:** neonatal hypoornithinemia and lethality rescuable by short-term arginine supplementation; postweaning hyperornithinemia; retinal degeneration in aged mice recapitulating the HOGA phenotype [66]	
Imerslund-Grasbeck syndrome 1 (**IGS1**)	261,100	Infancy- or early childhood-onset proteinuria and megaloblastic anemia due to vitamin B12 (cobalamin, Cbl) deficiency caused by vitamin B12 malabsorption	***Cubn*****KO:** no disease recapitulation, embryonic lethality [67]	
Infantile onset spinocerebellar ataxia (**IOSCA**)	271,245	Severe progressive neurodegenerative disorder characterized primarily by hypotonia, ataxia, ophthalmoplegia, hearing impairment, epilepsy and sensory axonal neuropathy	***C10orf2*****KI:** IOSCA mice manifest a mitochondrial epileptic encephalohepatopathy replicating the key findings of IOSCA patients [68,69]	Suitable model for testing metabolic interventions as treatment options for mitochondrial diseases
Lethal arthrogryposis with anterior horn cell disease (**LAAHD**)	611,890	Prenatal onset of diminished fetal mobility and contractures and postnatal respiratory failure resulting in perinatal death	** *Gle1* ** **KO (IMPC)**	
Lethal congenital contracture syndrome 1 (**LCCS1**)	253,310	A lethal condition of fetus with lack of movements, hydrops, micrognathia, pulmonary hypoplasia and multiple joint contractures	** *Gle1* ** **KO (IMPC)**	
Long-chain 3-hydroxyacyl-CoA dehydrogenase (**LCHAD**) deficiency *****	609,016	A mitochondrial disorder of long-chain fatty acid oxidation characterized by infancy- or early childhood-onset hypoglycemia, metabolic acidosis, hypotonia, liver disease, cardiomyopathy and arrhythmias, as well as a later onset of chronic peripheral neuropathy and pigmentary retinopathy	***Hadha*****KO:** embryonic lethality in homozygotes, hepatic steatosis at a young age (3 mo) and hepatocellular carcinoma without cirrhosis at an older age (>13 mo) in heterozygotes [70]	
Lysinuric protein intolerance (**LPI**)	222,700	Inborn error of amino acid metabolism resulting in growth failure, renal disease, hyperammonemia, pulmonary alveolar proteinosis, autoimmune disorders and osteoporosis	***Slc7a7*****KO1***:* growth restriction and very early embryonic lethality [71]	
			***Slc7a7*****KO2:** deletions do not recapitulate precisely the variants that have been reported in humans; key features of human LPI such as intrauterine growth restriction and proximal tubular dysfunction are present [72]	
Muscular dystrophy–dystroglycanopathy (congenital with brain and eye anomalies), type A, 3 (**MDDGA3**)	253,280	Brain and eye malformations, severe, congenital muscular dystrophy, mental retardation and survival up to more than 70 years	***Pomgnt1*****KO1:** viable mice developmental defects in muscle, eye and brain, similar to the phenotypes observed in humans [73,74] ***Pomgnt1* KO2**: increased postnatal lethality, mild dystrophy with reduction in muscle mass and muscle fibers and impaired muscle regeneration [75]	
Meckel syndrome type 1, (**MKS1**)	249,000	Genetically heterogenous disease with the main features being central nervous system malformation, polycystic kidneys, fibrotic changes in the liver, congenital heart malformation and polydactyly	***Mks1*****KO:** 259 amino acid deletion resulted in craniofacial defects, polydactyly, congenital heart defects, polycystic kidneys and randomized left-right patterning, quite similar to the human MKS1 phenotype [76]	
Mulibrey nanism (**MUL**)	253,250	Multi-organ disorder with prenatal onset growth failure, cardiomyopathy, characteristic craniofacial features, infertility, insulin resistance with type 2 diabetes and an increased risk for tumors	***Trim37*****KO:** recapitulates several features of the multi-organ human disorder, including infertility, increased risk for tumors, fatty liver and cardiomyopathy [77]	
Ovarian dysgenesis 1 (**ODG1**)	233,300	Hypergonadotropic hypogonadism with poorly developed streak ovaries in females and smaller testes and from low to normal sperm counts in men	***Fshr*****KO:** recapitulates human phenotype quite well; females have small ovaries due to a blockage of folliculogenesis, and male mice have smaller testes and reduced sperm counts [78,79,80]	
**PEHO** syndrome	260,565	Early infancy-onset hypotonia, delayed psychomotor development, infantile spasms, optic atrophy, progressive atrophy of the cerebellum and brainstem, dysmyelination and profound mental retardation	N/A	
Polycystic lipomembranous osteodysplasia with sclerosing leukoencephalopathy 1 (**PLOSL1**)	221,770	Adult-onset disorder of bones and central nervous system, leading to early dementia and death	***Tyrobp*****KO:** recapitulates skeletal and psychotic characteristics of PLOSL1 [81]	Promising drug therapy testing conducted using *Tyrobp* KO mice [82]
**RAPADILINO** syndrome	266,280	Radial and patellar aplasia, cleft or highly arched palate, diarrhea, dislocated joints, small size and limb malformations, long slender nose, cancer predisposition and normal intelligence	***Recql4*** ex 5-8 **KO:** embryonic lethality; ***Recql4*** ex 13 **KO**: neo- and postnatal lethality with growth retardation, skin, hair and bone defects; ***Recql4*** ex 9-13 **KO**: palate and limb defects and cancer predisposition [83,84,85]	
Retinoschisis (**RS1**)	312,700	X-linked childhood-onset reduced visual acuity due to retinal dystrophy leading to retinoschisis (splitting) of the neural retina in affected men	***Rs1*****KO:** not exactly recapitulating the human phenotype as disrupted organization of the retina was in all cell layers [86]	Successful gene replacement therapy in KO model [87], and dorzolamide treatment improved morphological features in 6 of 7 patients [88]
Salla disease (**SD**)	604,369	Hypotonia and delayed development in infancy, cerebellar ataxia, progressive cerebellar atrophy and dysmyelination leading to mental retardation; viscero- megaly and coarse features, enlarged lysosomes and high amounts of free sialic acid excreted in the urine	***Slc17a5* KO:** recapitulates hypomyelinating and lysosomal accumulation phenotype in CNS; in addition to premature death, poor coordination and seizures [89,90]	
Tibial muscular dystrophy (**TMD**)	600,334	Late adult-onset tibial muscular dystrophy	***Ttn*****KO:** embryonic or postnatal death; ***Ttn*** muscle-specific **KO:** adolescent death [91,92]	
Usher syndrome, type III (**USH3**)	276,902	Post-lingual, progressive hearing loss and loss of central visual acuity later in life	***Clrn1*****KO:** phenocopies early onset hearing loss, but not visual impairment [93,94]	Transgene strategy used for possible therapeutic intervention for Usher syndrome [94]

* Currently under consideration for FDH. KO (IMPC): mouse knockout has been generated by the International Mouse Knockout Consortium, and preliminary screening has been performed by the International Mouse Phenotyping Consortium (https://www.mousephenotype.org, accessed on 25 September 2021), but no scientific publication exists yet. CNS: central nervous system; KO: knockout; N/A: not available.

## Data Availability

Not applicable.

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
