# Peer review of "Modeling Rare Human Disorders in Mice: The Finnish Disease Heritage"

_cells, 2021, doi:10.3390/cells10113158_

Round 1

Reviewer 1 Report

The manuscript by Zárybnický and colleagues is a good review of the available murine models of pathologies included in the Finnish disease heritage panel.

The work is well written and easy to follow also from scholars from different fields. Nevertheless, I would suggest some modifications, as listed below.

  1. The title should better reflect the specificity of the review on FDH models.
  2. In Table 1, the last column is not very informative and sometimes misleading. For example, the "-" sign could indicate either that the mouse model poorly recapitulates the human disease or that the model is lacking. I would suggest using the column to name the mouse models (as done in Table 2), writing "N/A" when the model is lacking. In addition, it would be nice to read, in the same table, also the incidence of the disease.
  3. In Table 2:
    • COH1, HLS1, LAAHD, and LCCS1 syndromes: no indication of mice characteristics, despite the symbol "-" reported in Table 1;
    • The model for LCHAD (Hadha KO) is missing, although discussed later in the text;
    • instead of sometimes generic sentences such as "this model could be useful to study xyz..." the last column could be used to specify the differences and the limits of the model, reporting in the previous one only the characteristics recapitulating the human disease.
  4. It is not clear on which basis the diseases discussed in more detail in paragraph 2.1 have been chosen: are they the most frequent in the Finnish population, for example?
  5. Since no FHD models have been generated up to date with the CRISPR Cas9 method, the entire paragraph 3 is out of context and should be eliminated. It is enough the discussion of this genome editing method as it is in the Conclusions section.
  6. Finally, I'm not so sure that CRISPR Cas 9 is really the solution to reduce animal model use and meet all the 3Rs principle. I suggest less definitive statements on the subject.

Author Response

Reviewer 1:

The manuscript by Zárybnický and colleagues is a good review of the available murine models of pathologies included in the Finnish disease heritage panel.

The work is well written and easy to follow also from scholars from different fields. Nevertheless, I would suggest some modifications, as listed below.

  1. The title should better reflect the specificity of the review on FDH models.

We thank the reviewer for the valuable comments, which we find very useful for improving our manuscript. The title of our revised review manuscript is “Modelling rare human diseases disorders in mouse: the Finnish disease heritage”

  1. In Table 1, the last column is not very informative and sometimes misleading. For example, the "-" sign could indicate either that the mouse model poorly recapitulates the human disease or that the model is lacking. I would suggest using the column to name the mouse models (as done in Table 2), writing "N/A" when the model is lacking. In addition, it would be nice to read, in the same table, also the incidence of the disease.

Thanks for pointing out the difficulty to get the idea of last column of Table 1. As suggested, we have now included the mouse models (KO, cKO or KI) to each disease and replaced – with N/A.

We agree with the reviewer that for some readers it might be nice to include disease incidence to the table 1. However, reporting incidences is challenging because 1) they depend completely on the subpopulation and the geographic region of the country and 2) there is a lack of recent studies determining the incidence of several major mutations. Due to these reasons, we would not like to include disease incidences to our review.

  1. In Table 2:
    • COH1, HLS1, LAAHD, and LCCS1 syndromes: no indication of mice characteristics, despite the symbol "-" reported in Table 1;

Table 2 reports “Xy KO (IMPC)” for above mentioned diseases to indicate that the knockout has been generated by International Mouse Knockout Consortium and some preliminary phenotyping data may be present at International Mouse Phenotyping Consortium website but there is no scientific publication available. We have rephrased the footnote to better explain the situation in the revised manuscript.

    • The model for LCHAD (Hadha KO) is missing, although discussed later in the text;

We originally did not include LCHAD to the table 2 because it is not (yet) considered to fulfill the criteria of the Finnish disease heritage. LCHAD is now listed in the table 2 with a footnote indicating that it is currently under consideration for FDH.

    • instead of sometimes generic sentences such as "this model could be useful to study xyz..." the last column could be used to specify the differences and the limits of the model, reporting in the previous one only the characteristics recapitulating the human disease.

We agree with the reviewer that Note column in Table 2 previously included some generic information, which we have now removed. The column is also renamed as “Model utility” to better highlight our aim to include information about where the mouse model has been or might be very useful for therapeutic interventional studies.

  1. It is not clear on which basis the diseases discussed in more detail in paragraph 2.1 have been chosen: are they the most frequent in the Finnish population, for example?

We have better explained on the rows 119-123 of revised manuscript what is the basis of disease selection for their more specific description in 2.1

  1. Since no FHD models have been generated up to date with the CRISPR Cas9 method, the entire paragraph 3 is out of context and should be eliminated. It is enough the discussion of this genome editing method as it is in the Conclusions section.

As suggested by the reviewer, the entire chapter 3 is deleted from the revised manuscript.

  1. Finally, I'm not so sure that CRISPR Cas 9 is really the solution to reduce animal model use and meet all the 3Rs principle. I suggest less definitive statements on the subject.

We appreciate the reviewer’s comment on CRISPR technology in the context of 3R principles. We have toned down our statement and hope that the reviewer finds it appropriate now.

Reviewer 2 Report

The manuscript describes mouse models of 40 monogenic rare diseases that are enriched in the Finnish population and 19 defined as the Finnish disease heritage (FDH).

This review is well written and quite detailed and can interest distinct scientific communities.

I would propose to extend the perspective part related to humanization of mouse models. More details on the strategy to be followed with a possible schematic Figure can be a nice addition.

Author Response

Thank you for appreciating our review manuscript. As suggested, we now have extended the discussion about humanized mouse models as tools for rare disease studies and hope that the text on rows 467-479 together with the addition on rows 482-492 of the revised manuscript now meets your requirements.